# RB loss sensitizes cells to replication-associated DNA damage after PARP inhibition by trapping

Luis Gregory Zamalloa[1] , Margaret M Pruitt[1], Nicole M Hermance[1], Himabindu Gali[2], Rachel L Flynn[2], Amity L Manning[1]

The retinoblastoma tumor suppressor protein (RB) interacts physically and functionally with a number of epigenetic modifying enzymes to control transcriptional regulation, respond to replication stress, promote DNA damage response and repair, and regulate genome stability. To better understand how disruption of RB function impacts epigenetic regulation of genome stability and determine whether such changes represent exploitable weaknesses of RB-deficient cancer cells, we performed an imaging-based screen to identify epigenetic inhibitors that promote DNA damage and compromise the viability of RB-deficient cells. We found that loss of RB alone leads to high levels of replication-dependent poly-ADP ribosylation (PARylation) and that preventing PARylation by trapping PARP enzymes on chromatin enables RB-deficient cells to progress to mitosis with unresolved replication stress. These defects contribute to high levels of DNA damage and compromised cell viability. We demonstrate this sensitivity is conserved across a panel of drugs that target both PARP1 and PARP2 and can be suppressed by reexpression of the RB protein. Together, these data indicate that drugs that target PARP1 and PARP2 may be clinically relevant for RB-deficient cancers.

## Introduction

The retinoblastoma tumor suppressor (RB), which is lost or functionally inactivated in most cancers (Burkhart & Sage, 2008), is best known for its role as a negative regulator of E2F-dependent transcription (Dyson, 1998; Nevins, 2001). As RB becomes increasingly phosphorylated during G1 progression, its inhibition of E2F is abrogated, allowing for expression of key cell cycle genes, progression through the restriction point, and S-phase entry (Goodrich et al, 1991). However, the cellular function of RB is now appreciated to be much more extensive than E2F regulation (Velez-Cruz & Johnson, 2017; Dick et al, 2018) and proteomics analyses indicate that RB interacts with >300 proteins, nearly 20% of which are histones or epigenetic modifiers of histones (Sanidas et al, 2019).

Interactions with epigenetic regulators are believed to allow RB to orchestrate chromatin accessibility and transcription status across the cell cycle (Gonzalo & Blasco, 2005; Guzman et al, 2020). Reported RB interactors of this nature include histone acetylases (Manickavinayaham et al, 2019), deacetylases (Luo et al, 1998; Wang et al, 2019; Zhou et al, 2021), methylases, and demethylases (Nielsen et al, 2001; Vandel et al, 2001; Gonzalo et al, 2005; Blais et al, 2007; Chau et al, 2008; Ishak et al, 2016). Disruption of RB-dependent regulation of epigenetic factors compromises transcriptional regulation, induces replication stress, impairs DNA damage response and repair pathways, and corrupts genome stability (Manickavinayaham et al, 2020). These data suggest that epigenetic dysregulation may be a critical and exploitable feature of RB-deficient cells.

Indeed, recent reports demonstrate that RB-deficient cells are exquisitely sensitive to the inhibition of epigenetic modulators, including the key mitotic kinases Aurora A and Aurora B (Gong et al, 2019; Oser et al, 2019; Lyu et al, 2020; Yang et al, 2022). These kinases phosphorylate key regulators of centromere and spindle structure and function, and their inhibition synergizes with RB-dependent defects in centromere regulation and chromosome segregation (Iovino et al, 2006; Amato et al, 2009; Manning et al, 2010, 2014; Schvartzman et al, 2011). Other studies have described sensitivity of RB-deficient cells to pharmacological inhibition of polo-like kinase 1 function such that RB-proficient cells arrest in response to polo-like kinase 1 inhibition, whereas those deficient for RB continue to proliferate, accumulate high aneuploidy, and ultimately undergo cell death (Witkiewicz et al, 2018).

Prior studies have also implicated RB loss in conferring sensitivity to inhibition of poly ADP ribose polymerase (PARP) enzymes (Velez-Cruz et al, 2016; Jiang et al, 2020; Zoumpoulidou et al, 2021), yet the mechanistic explanation for this relationship remains unclear. Here, we identify PARP1/2 inhibitors in a screen for epigenetic modulators that exploit DNA damage phenotypes in cells lacking RB. We used isogeneic cell lines with and without RB to show that RB deficiency leads to high levels of replication-dependent PARylation. PAR, an epigenetic mark placed by PARP enzymes, marks sites of Okazaki fragment processing and replication stress. This modification functions to activate the DNA damage response pathways and stall replication until the stress is resolved (Amé et al, 2004; Sugimura et al, 2008; Bryant et al, 2009; Hanzlikova et al, 2018; Vaitsiankova et al, 2022). We find that when RB-deficient cells are

[1]Worcester Polytechnic Institute, Department of Biology and Biotechnology, Worcester, MA, USA    [2]Boston University School of Medicine, Pharmacology, Boston, MA, USA

Correspondence: almanning@wpi.edu

exposed to small molecules that lock PARP on chromatin, known as "trappers," cells progress into G2/M in the presence of unresolved DNA damage. This contributes to genomic instability and compromised cell viability. Restoration of RB expression in the isogenic cell line is sufficient to protect cells from replication stress associated with PARP trapping.

# Results

### RB-deficient cells are sensitive to PARP1 inhibition

Previous studies have found that RB plays a role in replication fork progression and homologous recombination (Marshall et al, 2019) and that RB-deficient cells are sensitive to DNA-damaging agents that generate double-strand breaks (Velez-Cruz et al, 2016; Aubry et al, 2020; Jiang et al, 2020). Regulation of the chromatin structure is critical to the repair of DNA damage and modulation of nucleosome positioning at sites of breaks is shown to be limiting for repair (Hauer & Gasser, 2017). Therefore, we hypothesized that defects in replication and homologous recombination that result from RB loss may render cells sensitive to epigenetic perturbations of chromatin structure. To test this possibility, we performed a targeted small molecule screen to assess measures of genome stability and viability after exposure to individual inhibitors from a panel of 96 small molecule modulators of epigenetic regulation. We designed a small molecule library to target 20+ categories of epigenetic modulators, including HDMs, HDACs, DNMTs, kinases, and PARPs (Table S1), that have been previously implicated in the maintenance of chromatin structure and genome stability (Putiri & Robertson, 2011; Sultanov et al, 2017; Karakaidos et al, 2020).

Using an hTERT RPE-1 (RPE) cell line in which RB can be depleted through doxycycline induced expression of an RB-targeting shRNA (Manning et al, 2014), cells were first depleted of RB for 48 h. Populations of cells with and without RB depletion were then exposed to individual epigenetic modulators for 48 h and assessed for early signs of sensitivity, including acquisition of DNA damage and reduced cell number. 17 inhibitors that caused significant death of control cells (defined as 80% or greater reduction in cell number after 2 cell cycles) were not evaluated further. To assess DNA damage after exposure to the remaining 79 inhibitors, cells were immunostained for γH2AX, a phosphorylated histone mark and early indicator of DNA double-strand breaks. Using Nikon Elements software, individual nuclei were identified by thresholding in the DAPI channel and sum intensity of γH2AX staining was measured (Fig S1A) to assess levels of DNA damage. Individual cells were considered to have enhanced DNA damage if nuclear γH2AX staining was more than twofold the average γH2AX intensity measured in control cells from the same screening plate. Relative fold change in the fraction of damaged cells was calculated to identify modulators that cooperate with RB loss to promote DNA damage. 14 inhibitors differentially induced a twofold or greater increase in DNA damage in cells depleted of RB with a robust Z-score of 3.0 or greater across experimental triplicates (Fig 1A and B).

Recent reports have indicated that RB-deficient cells are exquisitely sensitive to inhibition of Aurora kinase, EZH2 methyltransferase, and BRD4 inhibition (Ishak et al, 2016; Gong et al, 2019; Oser et al, 2019; Zhang et al, 2021). Consistent with these results, our

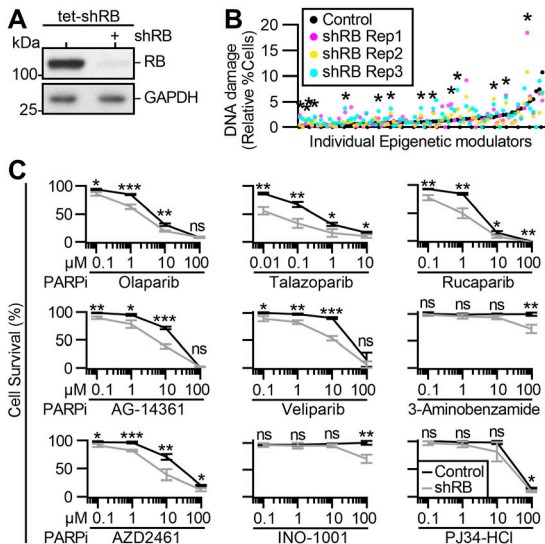

**Figure 1.  Identification of epigenetic sensitivities of RB-deficient cells.**
**(A)** Western blot analysis of cells with and without 96 h induced expression of an RB-targeting shRNA (tet-shRB). **(B)** Fraction of control (black) and shRB (magenta, yellow, and cyan) cells with DNA damage after 48 h of exposure to individual epigenetic modulators. * indicates differential increase in DNA damage in RB-deficient cells with robust Z score > 3. **(C)** Relative cell survival of control and shRB cells, as indicated by metabolic color conversion of PrestoBlue reagent after incubation with the indicated PARP inhibitors and concentrations. All experiments and statistics calculated between independent experiments performed in triplicate. Error bars represent SD between replicates. (*) $P < 0.05$; (**) $P < 0.01$; (***) $P < 0.001$; (ns) nonsignificant $P > 0.05$.

screen identifies inhibitors of Aurora kinase (via JNJ-7706621 and CYC116), inhibitors of EZH2 (via EPZ-6438 and 3-Deazaneplanocin A [DZNeP]), and inhibitors of BRD4 (via PFI-1 and Bromosporine) that each differentially enhance DNA damage levels in RB-depleted cells. Additional small molecules that scored as hits in our screen were inhibitors of HDAC (RGFP966, rocilinostat, and resminostat), JAK2 (ruxolitinib, AZD1480, and gandotinib), and PARP (olaparib and talazoparib). To further characterize these hits, we next assessed cell viability after exposure to a concentration range of each drug (Fig S1B). Of the 14 inhibitors assessed for impact on viability, only Olaparib and Talazoparib, both PARP1/2 inhibitors, differentially compromised viability of RB-deficient RPE cells in the 48–96 h time course of this experiment (Table S2).

Seven additional PARP inhibitors were represented in the screening library but did not meet the criteria described above to be considered hits on the screen. We therefore sought to determine if differential sensitivity may become apparent over a broader range of drug exposure. To this end, the viability of control and RB-depleted cells was assessed after 96 h of exposure to a concentration range of 0.01–100 $\mu M$ of each PARP inhibitor (Fig 1C). RB-deficient cells exhibited reduced viability compared with control cells after exposure to four inhibitors of PARP (rucaparib, veliparib, AG-14361, and AZD2461), but not to the remaining three (INO-1001, 3-aminobenzamide, and PJ34-HCl). The six inhibitors that selectively reduced viability of RB-depleted cells all target both PARP1 and PARP2, whereas the three that do not are described to selectively inhibit PARP1 but not PARP2 (Wahlberg et al, 2012; Ali et al, 2016). Interestingly, immunofluorescence-based analysis of γH2AX staining intensity

in control and RB-depleted cells verify that the PARP1/2 inhibitor rucaparib, but not veliparib, selectively promotes DNA damage in RB-deficient cells (Fig S1C and D). A key distinction between veliparib and rucaparib is the mechanism of action by which these drugs inhibit PARPs: rucaparib, like olaparib and talazoparib, perturb PARP function by trapping it on DNA (Zandarashvili et al, 2020). In contrast, veliparib is an enzymatic inhibitor of PARP function that does not trap the enzyme on DNA (Huang & Kraus, 2022). Together these data indicate that RB-deficient cells are generally sensitive to the combined inhibition of PARP1 and PARP2. These data additionally raise the possibility that the increased DNA damage seen in RB-deficient cells may stem from lesions caused not merely by loss of PARP1 and PARP2 functions, but from their sustained association with DNA when inhibited.

To validate PARP inhibition as an approach to specifically sensitize RB-deficient cells to high levels of DNA damage, we next measured γH2AX foci formation in a CRISPR-engineered *RB1* knockout cell line ([Nicolay et al, 2015]; RPE RB$^{KO}$) with and without RB re-introduction via an inducible Halo-tagged RB construct (RB-Halo; Figs 2A and B

and S2A and B). Cells were exposed to PARP inhibitors olaparib, rucaparib, talazoparib or veliparib for 48 h and analyzed for γH2AX foci. Here, γH2AX-positive DNA damage foci were quantified per nuclei and cells exhibiting greater than five foci were considered to have enhanced DNA damage. Similar to results from shRNA-mediated depletion of RB, RPE RB$^{KO}$ cells exhibit a differential increase in DNA damage after inhibition of PARP via olaparib, rucaparib or talazoparib, but not veliparib (Figs 2C and D and S2C). Critically, reintroduction of RB via ectopic expression of Halo-tagged RB construct (RB-Halo), but not of a Halo-tagged GFP construct (GFP-Halo), induced a partial decrease in the level of DNA damage in RPE RB$^{KO}$ cells exposed to olaparib, indicating that sensitivity to PARP trapping is dependent on loss of RB (Figs 2E and F and S2D and E).

## Accumulation of DNA damage after RB loss and PARP trapping is replication dependent

RB-deficient cells exhibit slow or stalled replication fork progression (Bester et al, 2011; Manning et al, 2014). The PARP enzymes

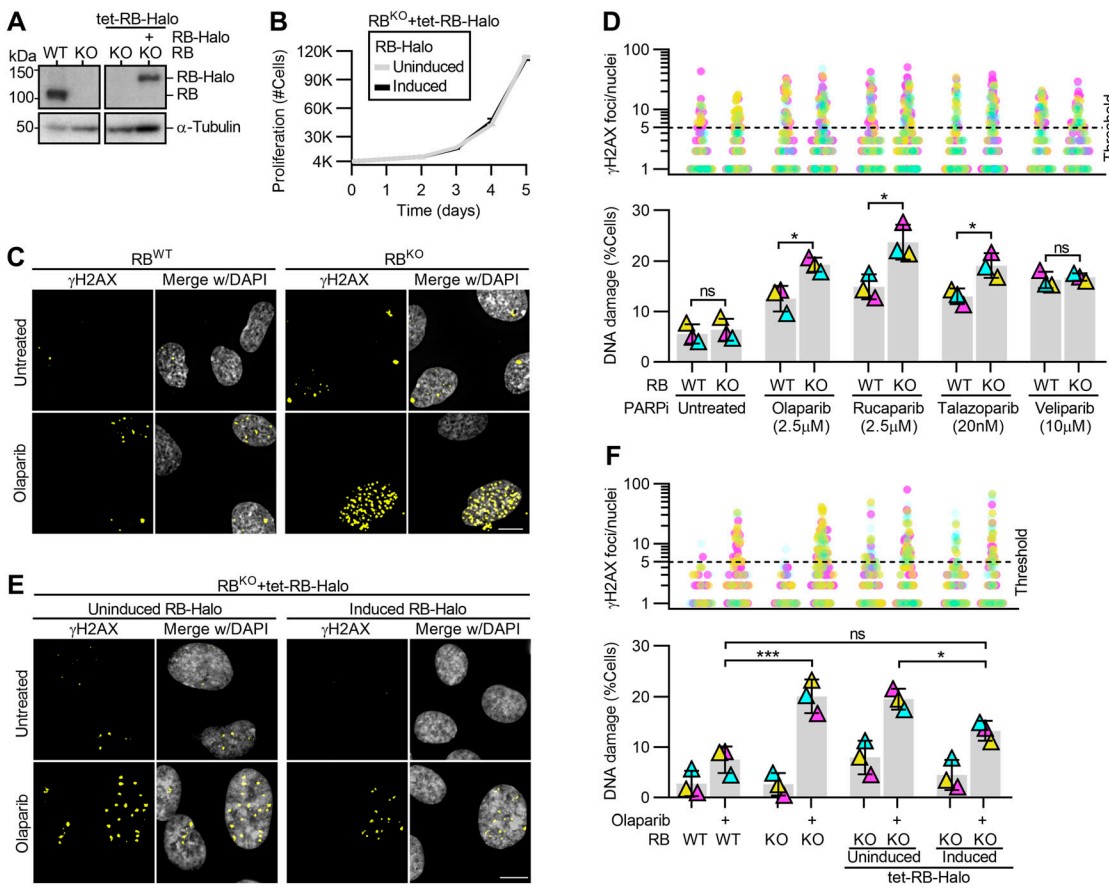

**Figure 2. PARP trapping sensitizes RB-deficient cells to high levels of DNA damage.**
**(A)** Western blot analysis of control (WT) and *RB1*-null (KO) RPE cells with and without doxycycline inducible RB-Halo (tet-RB-Halo). Cells were induced to express RB-Halo, as indicated (+). **(B)** Quantification of cell number in proliferative populations of RPE RB$^{KO}$ tet-RB-Halo cell number with and without 2 μg/ml doxycycline induction. **(C, D)** Representative images and quantification of γH2AX foci in RPE RB$^{WT}$ and RB$^{KO}$ cells with and without 48 h of incubation with the indicated PARP inhibitors. **(E, F)** Representative images and quantification of γH2AX foci in RPE RB$^{KO}$ tet-RB-Halo cells with and without doxycycline-induced RB-halo expression and after 48 h incubation with olaparib. **(D, F)** show the number of γH2AX foci per cell (top) and percent of cells with ≥5 damage foci (bottom). Scale bars are 10 μm. Experiments were performed and statistics calculated between independent experiments were performed in triplicate. Error bars represent SD between replicates. (*) *P* < 0.05; (**) *P* < 0.01; (ns) nonsignificant *P* > 0.05.

respond to replication stress by placing poly ADP ribose (PAR) modifications to initiate the DNA damage response (Amé et al, 2004) In addition, PARylation of substrates is necessary for efficient fork restart and Okazaki fragment processing during replication (Sugimura et al, 2008; Bryant et al, 2009; Hanzlikova et al, 2018; Hanzlikova & Caldecott, 2019; Vaitsiankova et al, 2022). These reports suggest that RB-deficient cells may rely on PARP-catalyzed PARylation to process their replication stress. To test this hypothesis, we examined cells in S-phase for evidence of PARylation. Control and RB-depleted cells were pulse-labeled with EdU for 30 min to enable identification of actively replicating cells, then fixed and immunostained for PAR. EdU was detected via click chemistry conjugation of a fluorophore, and DNA detected with DAPI. Intensity of PAR staining per nuclei was measured and compared between actively replicating cells in the control and RB-depleted populations. We find that S-phase cells exhibit an increase in nuclear PAR levels after RB depletion (Fig 3A and B). Similar results are seen if cells are treated with the PARG inhibitor PDD 00017273 (PARGi), to prevent turnover of PARylation marks (Hanzlikova et al, 2018; Vaitsiankova et al, 2022). Critically, inhibition of replication with emetine (Lukac et al, 2022), prevents the accumulation of PAR in RB-depleted S-phase cells (Fig 3A and B). These data indicate that during replication, RB-deficient cells experience stress that is sufficient to induce a PARP-dependent response.

If replication stress in RB-deficient cells underlies their sensitivity to PARP trappers, sites of DNA damage that follow PARP trapping should correspond with sites of replication stress. To test this possibility, we monitored both replication stress and accumulation of DNA damage concurrently. Replication protein A (RPA) binds to single-stranded DNA, and is phosphorylated at serine-33 (pRPA) in response to replication stress and replication-associated DNA damage (Marechal & Zou, 2015). Consistent with previous reports showing that RB loss promotes replication stress, we found a twofold increase in levels of pRPA staining when RB is depleted, compared with untreated RPE cells or mock-depleted RPE shRB cells. (Fig S3A and B; Bester et al, 2011; Manning et al, 2014). After 48 h of exposure to PARP trappers olaparib, rucaparib or talazoparib, we found a twofold to sixfold increase in the fraction of RB-depleted cells with 5 or more pRPA foci, compared with controls (Figs 4A and B and S3C and D). To confirm these results were specific to RB loss and not the result of potential off-target effects of the RB-targeting shRNA, we also analyzed pRPA and DNA damage foci after siRNA-mediated depletion of RB (Fig S4B–F) and in *RB1*-null osteosarcoma cells (Fig S4G–J). In both systems, we find results comparable with that described for shRNA-mediated depletion of RB: that RB loss sensitizes cells to high levels of both replication stress and DNA damage after PARP1/2 trapping. Notably, pRPA foci in RB-deficient, PARP1/2-trapped cells frequently co-localize with γH2AX foci (Fig S4A, F, and J), supporting a model whereby DNA damage is a consequence of unresolved replication stress.

To further define the extent to which RB loss sensitizes cells to PARP inhibition, we performed a sister chromatid exchange assay, differentially staining sister chromatids with acridine orange. In this assay, chromosome arm crossovers are a readout of homologous recombination-dependent repair of DNA breaks such that the

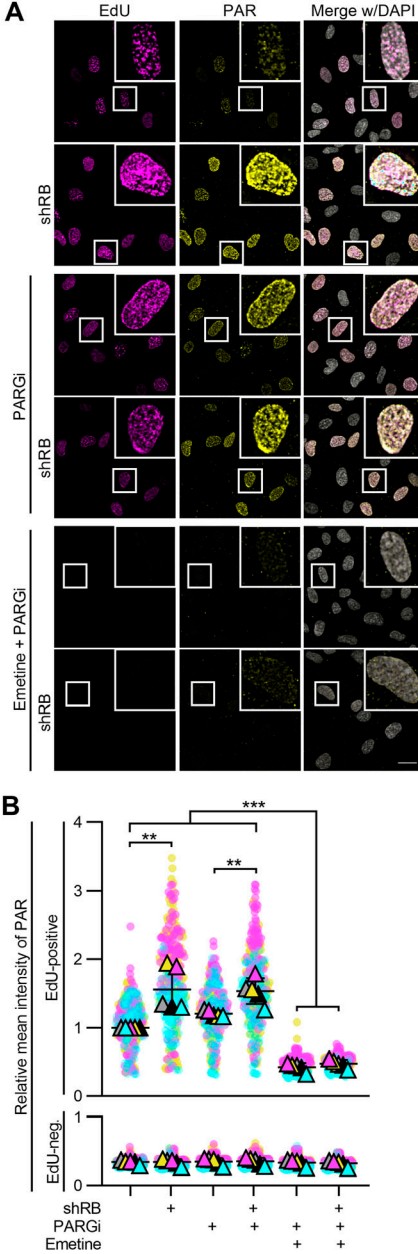

**Figure 3.  RB loss promotes replication-dependent PARylation.**
**(A, B)** Representative images and quantification of EdU and PAR intensity per cell for RPE tet-shRB cells with and without doxycycline induction of shRB (shRB). Cells were untreated or treated with 2 mM emetine for 1 h, and/or 10 μM PARG inhibitor for 30 min. All conditions were incubated with EdU for the final 30 min to label actively replicating cells. Scale bar is 20 μm. PAR intensity data are normalized to the EdU-positive, untreated RPE cells. Error bars represent SD and statistical analysis was performed between five independent experimental replicates. (**) $P < 0.01$; (***) $P < 0.001$.

number of crossovers indicate the frequency of DNA double-strand breaks in the preceding S/G2 phases of the cell cycle. We found that RB-depleted cells treated with the PARP trapper olaparib for 24 h display a significant increase in the number of crossovers per chromosome compared with controls (Fig 4C and D). Together, these data indicate that cells lacking RB are sensitive to the

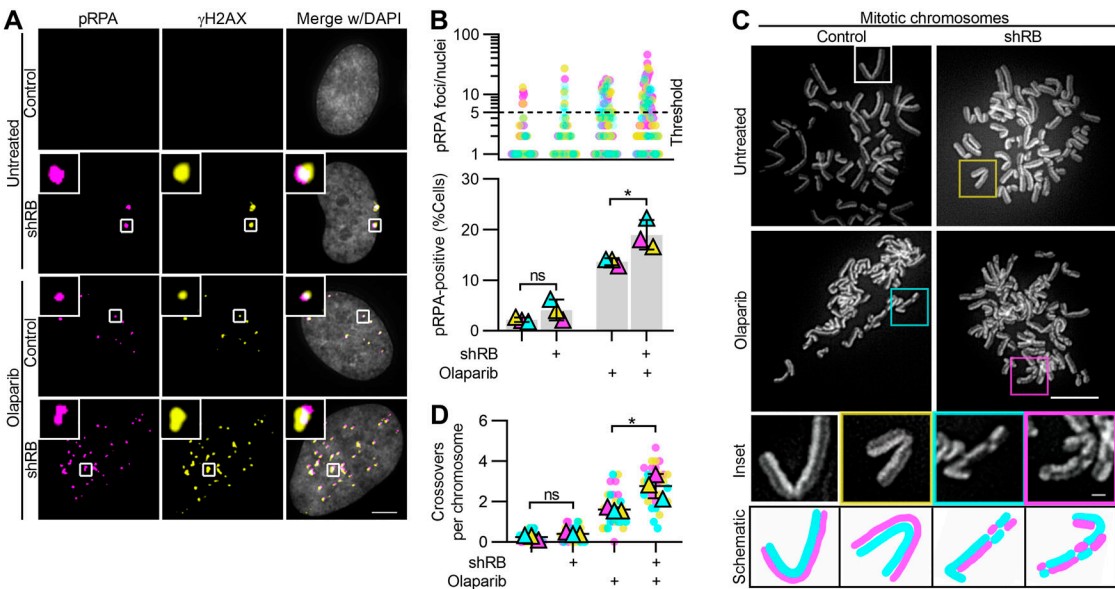

**Figure 4. DNA damage accumulates at sites of replication stress after combined PARP trapping and RB depletion.**
**(A, B)** Representative images and quantification of pRPA and γH2AX foci in RPE tet-shRB cells with (shRB) or without (control) doxycycline induction of shRB, after 48 h incubation with Olaparib, as indicated. Scale bar is 5 μm. **(B)** shows number of pRPA foci per cell (top) and percent of cells with ≥5 damage foci (bottom). Experiments were performed and statistics calculated between independent experiments performed in triplicate. Error bars represent SD between replicates. **(C, D)** Representative images and quantification of mitotic crossover events in RPE tet-shRB cells stained with acridine orange. Cells were cultured in 15 μM BrdU for 48 h and incubated with olaparib, as indicated, for the final 24 h before fixation. Scale bar is 10 μm. Insets represents 4x enlargements of a single chromosome, with 1 μm scale bar. Diagrams illustrate crossover events present in the inset. Independent experiments were performed in triplicate. Statistics were performed between the average number of crossover events per replicate (*) $P < 0.05$; (ns) nonsignificant $P > 0.05$.

acquisition of replication-dependent DNA double-strand breaks after PARP trapping.

## Persistent replication in the presence of damage perpetuates genomic instability and compromises cell viability

To examine the consequence of extensive DNA damage and investigate the possibility that cell cycle defects correspond with continued replication or translesion repair, cells were briefly pulsed with EdU, followed by examination of mitotic cells. RPE cells spend 2–4 h in G2 after completion of replication before mitotic entry. As a result, cells that enter mitosis during a 2-h EdU pulse are not expected to incorporate EdU. Consistent with this, only ~5% of control mitotic cells exhibit EdU foci after this short pulse. In contrast, ~25% of RB-depleted, PARP1/2-trapped cells display one or more EdU foci (Fig 5A and B). To distinguish whether EdU incorporation just before mitosis is a consequence of incomplete replication during S phase, or instead indicative of a shortened G2 phase, we repeated this analysis with a short, 30-min EdU pulse. Consistent with the longer EdU pulse, we find that RB-depleted cells exposed to PARP1/2 trappers, but not control cells, display an increased incidence of EdU foci during mitosis after a 30-min EdU pulse (Fig S5A and B). Given the short pulse of EdU, it is unlikely that this analysis is merely capturing the completion of normal S phase replication that precedes a shortened G2 phase. Instead, continued replication during or immediately before mitotic entry is indicative of an S phase exit and G2 progression with under-replicated DNA. Consistent with this interpretation, live cell imaging of an RPE

fluorescent, ubiquitination-based cell cycle indicator (FUCCI) cell line to monitor cell cycle progression indicated that duration of G2 is increased, not decreased, in RB-depleted Olaparib treated cells, compared with cells treated with only control or RB-targeting siRNA, or those treated with olaparib alone (Fig S5C and D).

Incompletely replicated DNA is susceptible to breaks when chromatin compacts in preparation for mitosis (Lezaja & Altmeyer, 2021). Indeed, we found that EdU foci in mitotic RB-depleted and PARP1/2-trapped cells frequently co-localize with γH2AX (Fig 5A and B). Consistent with the presence of chromatin breaks during mitosis, interphase RPE RB^KO cells treated with PARP inhibitors olaparib, rucaparib or talazoparib for 48 h exhibit a high frequency of micronuclei. RB loss alone has previously been shown to lead to whole chromosome segregation errors (Manning & Dyson, 2011). However, most of the micronuclei that result from the combined loss of RB and PARP1/2 trapping lack centromeres, indicating that the increase in micronuclei result from chromatin fragments that fail to incorporate into the main nucleus after mitotic exit, and not from an increase in whole chromosome segregation errors (Fig 5C and D).

Micronuclei are not only a consequence of genome instability but can serve to perpetuate further genomic lesions (Crasta et al, 2012; Soto et al, 2018). Therefore, to assess the long-term impact of increased genomic instability on RB-depleted cells, we monitored the replicative capacity of cells with and without PARP1/2 trapping and find that continued cell cycle progression cannot be maintained after both RB loss and PARP1/2 trapping. When incubated in media supplemented with EdU for 24 h, nearly 100% of control

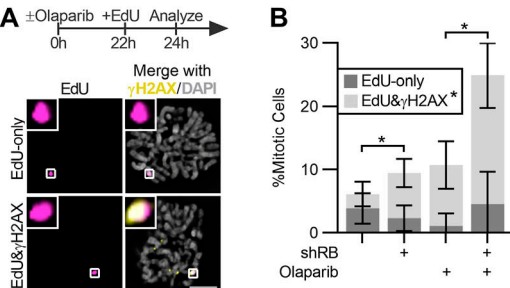

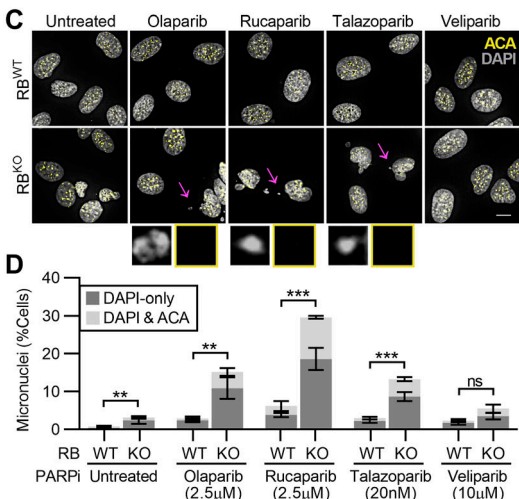

**Figure 5. Continued proliferation of RB-deficient cells in the presence of PARP trapping promotes genome instability.**
**(A, B)** Representative images and quantification of EdU and γH2AX foci in mitotic RPE tet-shRB cells with (shRB) or without (control) doxycycline induction of shRB, incubated with 2.5 μM olaparib for 24 h, and cultured in 10 μM EdU for 2 h before fixation. Images and analysis reflect analysis of mitotic cells. Scale bar is 10 μm. Error bars represent SD between three independent experimental replicates. Statistical analysis was performed for co-localized γH2AX and EdU foci, between replicates. **(C, D)** Representative images and quantification of the fraction of control (RB[WT]) and RB-null (RB[KO]) cells with micronuclei after incubation with the indicated PARP inhibitors for 48 h. Insets highlight micronuclei indicated by the arrow. Scale bar is 20 μm. Error bars represent SD between three independent experimental replicates and statistical analysis is performed between replicates. (*) $P < 0.05$; (**) $P < 0.01$; (***) $P < 0.001$; (ns) nonsignificant $P > 0.05$.

and RB-depleted cells, and over 80% of cells treated with PARP1/2 trapper alone, incorporate EdU, indicating that these populations are highly proliferative. In contrast, after 48 h of olaparib treatment, proliferation of RB-deficient cells is significantly compromised, with only ~40% of the cells within the population remaining competent to incorporate EdU (Fig 6A and B). Consistent with decreased replication, progression to mitosis is similarly reduced in RB-depleted cells in which PARP1/2 is trapped on chromatin (Fig 6C). To next determine if PARP1/2 trapping is cytotoxic for RB-deficient cells, or merely cytostatic, we assessed populations of control and RB-depleted cells for cell death after olaparib treatment. PARP1/2 inhibition impairs caspase-dependent mechanisms of cell death (Zhang et al, 2012; Tsikarishvili et al, 2021) making readouts of apoptosis ineffective for this system. We therefore opted for live cell imaging to assess the frequency at which cells exhibit blebbing and

loss of anchorage, indicative of cell death. Consistent with viability assays (Fig 1C), live cell imaging confirms that concentrations of PARP1/2 inhibitor that are sublethal for control cells, increase death of RB-depleted and RPE RB[KO] cells within 72 h (Fig 6D–F). Similar results are seen for RB-null SAOS2 osteosarcoma cells where dead/dying cells are not apparent in control populations but become prevalent after PARP1/2 trapping (Fig 6G and H).

## Discussion

Together, this study defines an increased sensitivity of RB-deficient cells to trapping of PARP1/2 enzymes on chromatin and mechanistically links this phenotype with replication-induced DNA damage. We show that RB-deficient cells experience stress during replication that promotes robust and sustained PARylation. After exposure to PARP1/2-trapping drugs, DNA lesions that would otherwise slow replication in a PAR-dependent manner appear to be bypassed as cells progress through the S phase. Using a combination of acute depletion of RB and constitutive *RB1* knockout, we show that when PARP1/2 is trapped on DNA sites of replication stress accumulate DNA damage during S phase and that persistent or incomplete replication during mitotic chromatin compaction sensitizes G2/M cells to further DNA damage. The resulting genome instability is evident in fragmented chromatin that accumulate in micronuclei in subsequent cell cycles. In line with previous studies that show RB loss is synthetic lethal with PARP inhibition in osteosarcoma cells (Velez-Cruz et al, 2016; Jiang et al, 2020; Zoumpoulidou et al, 2021), we find that these assaults to genomic integrity correspond with reduced proliferation and increased cell death both in contexts where RB is experimentally depleted, and in cancer cells where the RB1 gene is deleted. These data posit that defects in RB-deficient cells' capacity to restrain replication in response to stress and/or damage may underlie this synergy. Together with work by Mittnacht and colleagues (Zoumpoulidou et al, 2021), our data provide evidence that RB status is a clinically relevant biomarker for selection of PARP inhibitors and suggest that RB-deficient cells may be similarly sensitive to additional modulators of replication stress and fork restart (Witkiewicz et al, 2018; Ubhi & Brown, 2019).

Of the nine PARP inhibitors evaluated in this study, inhibitors targeting both PARP1 and PARP2, but not those specific for PARP1, had a differential impact on RB-deficient cell viability, suggesting that in the context of this study, PARP1 and PARP2 have redundant roles. This is consistent with reports implicating both PARP1 and PARP2 as early sensors of DNA damage and the first line of defense against genomic instability (Hottiger, 2015). PARP is activated by binding to sites of single-stranded DNA (Yang et al, 2004), initiating DNA damage repair pathways (Okano et al, 2003). In addition, PARP1 and PARP2 have important roles in regulating base excision repair (Ronson et al, 2018) pathways and non-homologous end joining (NHEJ)-based repair of DNA double-strand breaks (Luijsterburg et al, 2016). This role in NHEJ makes both PARP1 and PARP2 clinically relevant targets in tumors where the complementary double-strand break repair

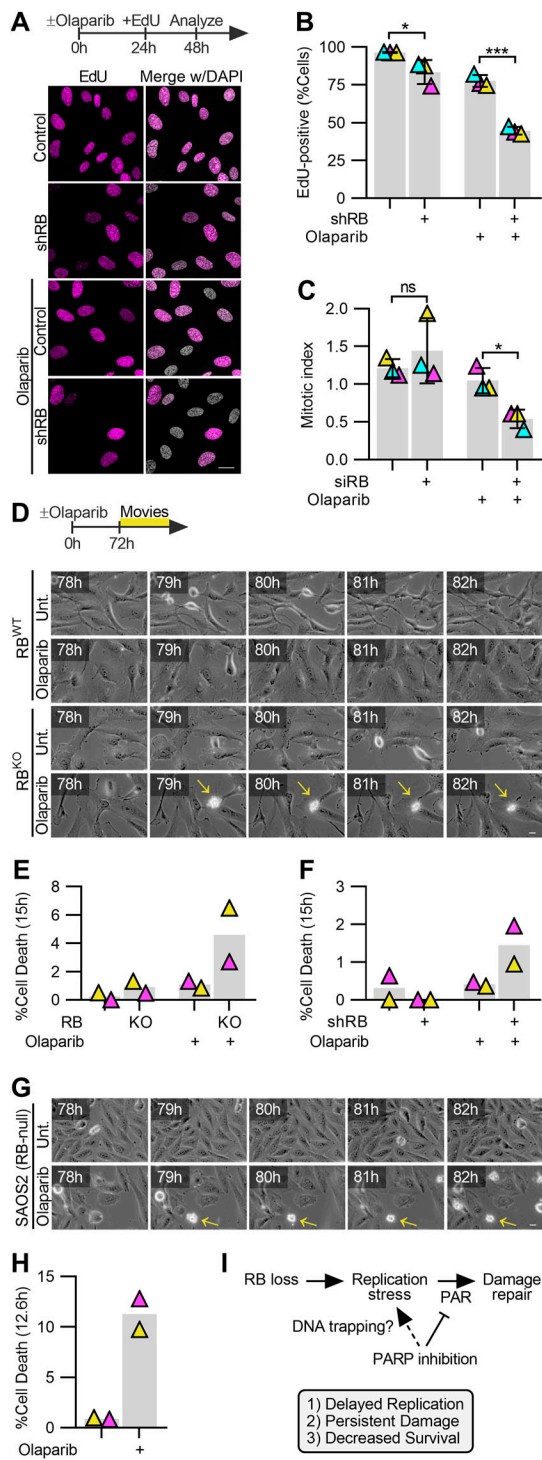

Figure 6. **PARP trapping synergizes with RB loss to compromise cell viability.**
**(A, B)** Representative images and quantification of replication-competent RPE tet-shRB cells with (shRB) or without (control) doxycycline induction of shRB, after incubation with 2.5 μM olaparib for 48 h. Cells were incubated with 10 μM EdU for the final 24 h before fixation. Scale bar is 20 μm. **(C)** Quantification of the fraction of RPE cell populations in mitosis after siRNA-based depletion of RB, then incubation in 2.5 μM olaparib for 48 h. Error bars represent the SD between three independent experimental replicates. Statistical analysis was performed between replicates. **(D, E, F, G, H)** Representative still frames and quantification of cell death from live cell imaging of control (RB^WT) and RB-null (RB^KO) cells, (F) in

pathway, homologous recombination (HR) is already compromised (Bryant et al, 2005; Farmer et al, 2005). Given the previously described role for RB in HR-dependent DNA damage repair pathways (Marshall et al, 2019), it is tempting to speculate that synergy between RB loss and PARP inhibition occurs as a result of dual inhibition of HR and NHEJ. However, this model fails to account for the persistent replication during G2/M that is apparent when both RB and PARP1/2 functions are compromised, but not when either is lost alone. Instead, our findings are consistent with a model whereby PAR-dependent signaling is necessary to sustain fork stability and prevent replication fork progression when DNA lesions are present. In the absence of PAR, RB-deficient cells may bypass such replication impediments, resulting in under-replicated, single-stranded regions of DNA that recruit pRPA and are the sites of continued replication late in G2/M. In addition, our observation that inhibitors functioning to trap PARP on chromatin, but not veliparib which inhibits poly ADP ribosylation without trapping, enhance DNA damage in RB-deficient cells suggesting that robustness of this synergy may require loss of poly ADP ribosylation and be exacerbated by the physical impediment to replication that results from PARP trapping. This model (Fig 6I) is supported by the replication-dependent nature of pRPA and γH2AX accumulation and evidence of under-replicated regions of DNA in mitotic cells in which both RB and PARP functions are compromised. This model is also consistent with reports demonstrating that RB loss facilitates translesion synthesis in the presence of crosslinking agents (Bosco et al, 2004) and that RB-null cancers are sensitive to loss of a number of DNA damage repair proteins (Aubry et al, 2020).

## RB loss as a biomarker to predict sensitivity to epigenetic perturbation of replication stress response

Loss or functional inactivation of the retinoblastoma tumor suppressor protein RB is common in a variety of human cancers (Burkhart & Sage, 2008; Peifer et al, 2012; George et al, 2015; Sanchez-Vega et al, 2018). Genomics (Burkhart & Sage, 2008; George et al, 2015) and transcriptomics (Chen et al, 2019) analysis of cancer patients reveal that *RB1* alterations predict poor clinical outcomes, raising the need for targeted therapies. Synthetic lethality is a phenomenon where alterations in one gene hypersensitize cells to alterations in another gene by means of pathway dependence or redundancy, and it presents an exciting avenue to target RB-deficient cancers. Recent reports describe synthetic lethal interactions of RB-deficient cells with epigenetic modulators including inhibitors of Aurora kinases (Gong et al, 2019; Oser et al, 2019; Yang et al, 2022) and EZH2 (Ishak et al, 2016). Our data add to a growing body of work showing that RB-deficient cells are exquisitely sensitive to PARP trapping.

RPE tet-shRB cells with (shRB) or without (control) doxycycline induction of shRB, and, (G, H) *RB1*-null SAOS2 osteosarcoma cells incubated with or without olaparib. Scale bars are 20 μm. Timestamps indicate time since PARP inhibitor addition. Live cell imaging was performed in duplicate. (*) *P* < 0.05; (***) *P* < 0.001; (ns) nonsignificant *P* > 0.05. **(I)** model of the proposed mechanism underlying replication stress, DNA damage, and reduced viability in Rb-deficient cells after PARP trapping.

## Materials and Methods

### Cell culture, transfection, and immunofluorescence

hTERT RPE-1 FUCCI cells were cultured in DMEM/F12 (GenClone) and other RPE-1 cell derivatives were cultured in DMEM (GenClone) supplemented with 10% FBS (Sigma-Aldrich) and 1% penicillin/streptomycin (Gibco). RPE FUCCI cells were kindly provided by Dr. Neul Ganem (Boston University School of Medicine) RPE $RB1^{KO}$ and parental RPE cells were kindly provided by Dr. Nick Dyson (MGH Cancer Center). SAOS-2 cells were cultured in McCoy's 5A Medium (Gibco) supplemented with 15% FBS (Sigma-Aldrich) and 1% penicillin/streptomycin (Gibco). All cells were maintained at 37°C and 5% $CO_2$. Acute RB depletion was obtained through addition of 2 $\mu$g/ml doxycycline to induce expression of an RB-targeting shRNA (Manning & Dyson, 2011), or alternatively via reverse-transfection of a pool of four RB-targeting siRNA sequences for 48 h, as previously described (Manning et al, 2014). For all experiments, siRNA- or shRNA-driven RB depletion was performed 48 h before treatment with inhibitors to PARP or other epigenetic modulators. RPE $RB1^{KO}$ tet-RB-Halo cells were generated via lentiviral transduction of a pLenti CMV/TO $RB1$-Halo construct. Depletion and/or induced expression of RB was monitored by Western blot analysis of whole cell lysates prepared using 2x Laemmli sample buffer (#1610737; Bio-Rad) supplemented with 2-mercapto-ethanol (#BP176-100, 1:20; Thermo Fisher Scientific). Further information on reagents and antibodies used can be found in Table S3.

For the epigenetic modulator, screen cells were cultured in imaging bottom dishes (#3904; Corning) and incubated with individual drugs at a final concentration of 10 $\mu$M for 48 h before fixation in 4% PFA and $\gamma$H2AX immunofluorescence as previously described (Manning et al, 2014). Cells were subsequently counterstained with fluorophore-conjugated secondary antibodies (Invitrogen) and 0.2 $\mu$g/ml DAPI. 10 mg/ml DABCO (#112470250; Thermo Fisher Scientific) in glycerol-PBS (9:1) antifade reagent was used to stabilize the signal. $\gamma$H2AX and/or pRPA immunofluorescence was performed in cells grown on coverslips and fixed in 4% PFA, as previously described (Manning et al, 2014). Replicating cells were labeled with a pulse of 10 $\mu$M EdU for 2–24 h, as indicated in the text, and visualized using the Click-iT EdU Imaging Kit (#C10637; Invitrogen), as per the manufacturer's instructions. To detect PAR, cells were processed as previously described (Vaitsiankova et al, 2022) and incubated with PAR antibody. Cells were counterstained with fluorophore-conjugated secondary antibodies and 0.2 $\mu$g/ml DAPI, mounted on slides with ProLong Gold, and imaged using a Nikon Eclipse Ti-E microscope equipped with a Zyla sCMOS camera and controlled by NIS-Elements software. All experiments were performed in triplicate and a minimum of 30 cells per condition were analyzed. Experimental conditions from within a biological replicate were imaged in parallel and at the same exposure time. Representative images were deconvolved in NIS-Elements software.

For live cell imaging, cells were grown on 12-well plates and treated with olaparib for 72 h. Cells were imaged using the Nikon Eclipse Ti-E equipped with perfect focus software and an environmental chamber to maintain 37°C and 5% $CO_2$. Phase contrast images were captured every 5 min for the final 12–15 h of olaparib incubation. Gamma adjustments were made in the phase contrast channel of representative movie stills to enable better visualization across time.

### Cell proliferation and viability assays

For each of three independently prepared and analyzed replicates, cells were incubated with the indicated inhibitor concentrations in technical duplicates. At 96 h, cells were suspended and manually counted with a hemocytometer. Alternatively, PrestoBlue Cell Viability Reagent (#A13261, 1:10 dilution; Invitrogen) was added at indicated timepoints and incubated for 3 h. Fluorescence was analyzed at 560/590 nm ex/em with a PerkinElmer Victor3 1420 plate reader.

### Sister chromatid exchange assay

To differentially label replicated sister chromosomes, cells were incubated in 15 $\mu$M BrdU for 48 h (~2 cell cycles). When indicated, olaparib was added for the final 24 h of BrdU labeling. Chromosome spreads were prepared by treating cells with 0.1 $\mu$g/ml Nocodazole for 30 min to depolymerize microtubules, followed by incubation in 75 mM KCl for 16 min. Cells were fixed in methanol–acetic acid (3:1) for 20 min at 4°C. Fixed cells were then dropped onto slides, dried overnight in the dark, stained with 100 $\mu$g/ml acridine orange (#A3568; Molecular Probes), and mounted in 0.1 M $Na_2HPO_4$ and 0.1 M $KH_2PO_4$.

### Image analysis and statistics

For the drug sensitivity screen, extended depth of focus 2D-projections were generated using NIS-Elements software. Nuclear regions were identified based on a DAPI threshold and $\gamma$H2AX staining intensities were measured. Cells were considered damaged if the sum intensity of a given cell was twofold or greater than the average intensity observed in the untreated controls. For imaging of cells on coverslips, a Cell Profiler (Stirling et al, 2021) pipeline was generated to first identify nuclei and then detect individual $\gamma$H2AX-positive foci. Cells were considered damaged if the $\gamma$H2AX foci count per nuclei was 5 or greater. Thresholds for nuclear EdU staining intensities were set on a per replicate basis and kept consistent across conditions within an individual replicate. SuperPlots represent 100 cells scored per experimental condition and triangles indicate averages per biological replicate. Statistical analysis was performed across independent experimental triplicates. Where relevant, technical replicates were averaged before comparisons were made between biological triplicates.

Relative $IC_{50}$ calculations for Table S2 were performed in GraphPad Prism software, using four-parameter logistic regression with iterative predictions of the following equation:

$$Y = Bottom + (Top - Bottom)/(1 + 10\verb|^|((LogIC50 - X) \times HillSlope)),$$

where X is a concentration ($\mu M$) tested for a given drug, Y is the respective % Cell survival, Top and Bottom are plateaus in units of % Cell survival, HillSlope indicates the steepness of the curve, and LogIC50 indicates the logarithm in base 10 of the concentration required to bring the curve down to the point halfway between Top and Bottom.

Unless stated otherwise, statistical analyses are two-tailed unpaired $t$ tests, with $P$-values indicated in figure legends. For the screen statistical analysis, robust-Z-scores were calculated as previously described (Chung et al, 2008) via the following equation:

$$MAD = 1.4826 \times median\left(\left|x_{ij} - median(x)\right|\right),$$

where $x_{ij}$ indicates the fold change of percent damaged cells between RB-depleted and control cells for a given experimental condition (shown in Table S1), and x indicates the median among the 79 epigenetic modulators tested. Robust Z-scores are defined by $x_{ij}$ divided by MAD. Hit discovery was assessed with a threshold of $Z \geq 3$.

## Supplementary Information

## Acknowledgements

We thank members of the Manning and Flynn Labs for technical assistance and critical reading of the article. This work was supported by a National Institute of Health award R01CA201446 to RL Flynn and American Cancer Society RSG-21-066-01- CCG and Smith Family Awards to AL Manning.

### Author Contributions

LG Zamalloa: data curation, formal analysis, validation, investigation, visualization, methodology, and writing—original draft, review, and editing.
MM Pruitt: data curation, formal analysis, validation, investigation, visualization, methodology, and writing—review and editing.
NM Hermance: data curation, formal analysis, investigation, methodology, and writing—review and editing.
H Gali: data curation, formal analysis, investigation, and writing—review and editing.
RL Flynn: conceptualization, supervision, funding acquisition, visualization, methodology, and writing—review and editing.
AL Manning: conceptualization, data curation, formal analysis, supervision, funding acquisition, validation, investigation, visualization, methodology, and writing—original draft, review, and editing.

### Conflict of Interest Statement

The authors declare that they have no conflict of interest.

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
