## [Reviewer comments · Life Science Alliance]

Life Science Alliance

RB loss sensitizes cells to replication-associated DNA damage following PARP inhibition by trapping

Luis Zamalloa, Margaret Pruitt, Nicole Hermance, Himabindu Gali, Rachel Flynn, and Amity Manning

DOI: <https://doi.org/10.26508/lsa.202302067>

Corresponding author(s): Amity Manning, Worcester Polytechnic Institute

Review Timeline:

Submission Date:	2023-03-31
Editorial Decision:	2023-05-08
Revision Received:	2023-08-04
Editorial Decision:	2023-08-29
Revision Received:	2023-09-05
Accepted:	2023-09-06

Transaction Report:

May 8, 2023

Re: Life Science Alliance manuscript #LSA-2023-02067-T

Amity Manning
Worcester Polytechnic Institute
Biology and Biotechnology
60 Prescott St
Worcester, MA 01605

Dear Dr. Manning,

Thank you for submitting your manuscript entitled "RB loss sensitizes cells to replication-associated DNA damage following PARP inhibition" to Life Science Alliance. The manuscript was assessed by expert reviewers, whose comments are appended to this letter. We invite you to submit a revised manuscript addressing the Reviewer comments.

Thank you for this interesting contribution to Life Science Alliance. We are looking forward to receiving your revised manuscript.

Sincerely,

B. MANUSCRIPT ORGANIZATION AND FORMATTING:

Reviewer #1 (Comments to the Authors (Required)):

This study conducted a targeted small molecule screen to investigate the sensitivity of RB-deficient cells to inhibitors of epigenetic regulators. The results showed that RB loss renders cells sensitive to PARP1/2 inhibitors, which compromise cell viability. Overall, the study suggests that defects in replication and homologous recombination resulting from RB loss may render cells sensitive to epigenetic perturbations of chromatin structure. RB loss sensitizes cells to high levels of both replication stress and DNA damage following PARP1/2 inhibition, and cells lacking RB are sensitive to the acquisition of replication-dependent DNA double strand breaks. The study also suggests that continued cell cycle progression cannot be maintained following both RB loss and PARP1/2 inhibition.

Point 1: RB-deficient cells are sensitive to PARP1 inhibition - strongly supportive

Point 2: Accumulation of DNA damage following RB loss and PARP inhibition is replication dependent - supportive

Point 3: Persistent replication in the presence of damage perpetuates genomic instability and compromises cell viability - very supportive

Expression of shRNA and RB in the systems used is induced by doxycycline. The effect of doxycycline appears to be uncontrolled for in these experiments. Similarly, the shRNA vector used for the doxycycline-inducible system is uncontrolled for using scramble shRNA or tet-Halo for the tet-RB-Halo.

"Together these data demonstrate that RB-deficient cells are generally sensitive to combined inhibition of PARP1 and PARP2" is technically inaccurate given the author's results showing that Veliparib failed to affect cell survival. What renders sensitivity to PARP "inhibitors" is the trapping of PARP rather than inhibition of PARP function. Please revise the statement.

Following with the previous comment, the discussion fails to address this observation. If PAR-dependent signaling is necessary to sustain fork stability, how do the authors explain the absence of effects upon Veliparib treatment?

"Critically, re-introduction of RB... is sufficient to rescue DNA damage..." Figure 2F graph is missing the comparison between WT+Olaparib and tet-RB-Halo+Olaparib that is relevant for this statement.

The increased in EdU foci presented in Figure 5 could also be interpreted as RB-depleted cells having shorter G2 phase following PARP trapping, yet this was dismissed. Cell cycle analysis of these cells would strengthen the conclusion that the increased EdU foci are indicative of under-replicated DNA.

Minor:

Figure 1 legend: panel B indicates that control is in black and shRB in gray, but that is the color scheme for panel C. Panel B has different colors for each shRB repeat.

Several figure legends: "biological replicates" unless these are 5 distinct cell lines, this are experimental replicates.

Reviewer #2 (Comments to the Authors (Required)):

This research conducted by Zamalloa et al., sought to identify a mechanistic explanation to rationalize previous research that demonstrated loss of the tumor suppressor retinoblastoma (RB) compromises genomic stability, renders cells sensitive to inhibition of epigenetic modulators, and confers sensitivity to inhibition of poly ADP ribose polymerase (PARP) enzymes. Screens using inhibitors to epigenetic regulatory enzymes identified a group of inhibitors that selectively reduced viability of RB-depleted cells through targeting of PARP1/2. The authors were able to identify the specific mechanism of reduced viability through comparison of drug mechanisms related to PARP inhibition. Controlled knockdown and re-expression of RB in cells receiving Olaparib treatment identified PARP1/2 combined inhibition sensitizes RB-deficient cells to high levels of DNA damage,

and re-expression of RB was sufficient to reduce DNA damage in response to PARP1/2 inhibition. Figure 3 experiments used quantification of Edu and PAR following pulse-labelling clearly identified RB loss promotes replication dependent PARylation. Combined with the findings of Figure 4, which identified DNA damage accumulation at sites of replicative stress following PARP1/2 inhibition and RB depletion, these two figures provide strong supporting evidence for their conclusion that accumulation of DNA damage following RB loss and PARP inhibition is replication dependent. The extended effects of persistent replication in the presence of this damage were also investigated and the researchers concluded an increase in under-replicated DNA, increased frequency of micronuclei, and continued cell cycle progression cannot be maintained following RB loss and PARP inhibition.

Overall, this paper does well in connecting findings of previous research that identified RB and its role in genomic stability and how it can impact therapeutics through a specific and well-described mechanism. This paper provides clear mechanistic groundwork that would support more translational based investigations, for example, if these effects could be reproduced in various cancer cell lines and if there is synergistic or contraindicatory effects with other cancer therapeutics. However, upon review there are some potential revisions to be considered by these researchers.

1) The representative images of Figure 5C and their insets are difficult for the reader to identify the presence of micronuclei from the given images. The current magnification box partially obscures the identified cell in question. Recommend this be revised and that more levels of magnification, and some images without annotations to allow readers to see the micronuclei for themselves.

2) The second recommendation is the addition of a schematic summary in Figure 6 or as an additional figure. This is a largely mechanistic study, so providing a clear description of the postulated mechanism of how RB loss cooperates with PARP1/2 inhibition is key.

3) The final recommendation would be review of minor editing in the figure legends and text. For example, the description of Figure 1B states this graph should be made of black and grey data points where it is made of four different colored data points.

Referee Cross-Comments - The other reviewers have questions about the complementation experiment where RB is re-expressed in RB knock out cells. This is clearly a partial effect. However, compared to other attempts at this type of RB add back made over the years it is quite impressive. Likely because RB regulates so many different epigenetic mechanisms at different stages of the cell cycle, we've all discovered that you can't re-program cells back to normal with brief re-expression of RB. I see this experiment as a strength.

Reviewer #3 (Comments to the Authors (Required)):

The manuscript reports selectively increased sensitivity to PARP1,2 inhibitors, independently confirming reports from several preceding publications, but expanding these observations to an engineered retinal cell model. Interestingly, and notably, the authors show data that RB1 loss leads to increased PARylation in S-phase cells, raising the possibility that increased reliance on PARylation during S-phase explains the increased PARP inhibitor sensitivity of cells lacking RB.

Hence, the manuscript presents novel and attractive information. However, there are shortcomings in this manuscript that the authors should seek to address. On many occasions, the authors seem to jump to conclusions and a balanced view of the data is not presented. Most significantly, selective sensitivity in their cell model is seen using Veliparib, albeit this PARP inhibitor does not show the described increase in pRPA and associated DNA damage phenotype, raising the question of whether these additional phenotypes, which are extensively studied in the work, are relevant for the increased sensitivity seen in their cell model.

The veliparib data call into question statements in the abstract and throughout the manuscript whereby "PARP inhibition" acts by causing replication-associated DNA damage. It is likely that many of the observations made are a consequence of the PARP complex trapping, and do not reflect PARP inhibition. How the observed responses relate to the selective sensitivity to these inhibitors is unclear. As a minimum these concerns need to be addressed by robustly rewriting the manuscript, detailing that the observed responses are not seen with Veliparib and ideally covering this discrepancy, and its implications, in the discussion. The abstract should be carefully examined. A number of statements presented are challenged by experimental data using Veliparib.

Other concerns that the authors should seek to address are:

There appear to be a number of unsupported statements throughout the manuscript.

For example:

The introduction states: "we find that inhibition of PARP activity permits acceleration of replication fork progression" albeit no experiments are shown that measure fork progression.

The result section states: "RB1 loss sensitises cells to high level of both replication stress" ... while in fact, the data show no difference in pRPA (a common readout for replication stress) between RB-containing and RB-depleted cells (figure 4B, supplemental figure 3B supplemental figure 4D). Hence at least using this canonical assay for replication stress there is no evidence supporting the statement made.

The discussion states that: "We show .. that when PARP1/2 is inhibited sites of replication stress accumulate DNA damage during S phase...". This statement is contradicted by data using Veliparib, which inhibits PARP 1/2 but does not cause either of

these phenotypes

that:

further down is a statement: "PARP is activated by single strand DNA, parylating various substrates to stabilize the replication fork (Yang et al 2004)", yet the reference cited does not report any such data.

statistics use:

The method section states that unpaired t-tests were used to assess the significance of variance. However, this is not meaningful and appropriate for some data. Specifically, a 2-way ANOVA scoring for "variables interaction" should be used to assess the ability of RB re-expression to rescue the DNA damage following Olaparib. The same applies to data in Figure 6B assessing if Olaparib "interacts" with RB1 loss to decrease EDU-positive cells.

Minor :

The concentration of all drugs used should be detailed in the respective figure legends.

Data volume statements, please can it be clearly spelled out for each dataset if biological repeats for experiments were run in parallel, or were independent.

We thank the reviewers and Editor for careful review and thoughtful feedback on our manuscript. We have addressed each critique, as described below. We believe that the experimental additions and textual clarifications have increased the robustness and clarity of the manuscript and we respectfully request that you consider it for publication.

Point by point response to reviewers' critiques:

Reviewer #1 (Comments to the Authors (Required)):

This study conducted a targeted small molecule screen to investigate the sensitivity of RB-deficient cells to inhibitors of epigenetic regulators. The results showed that RB loss renders cells sensitive to PARP1/2 inhibitors, which compromise cell viability. Overall, the study suggests that defects in replication and homologous recombination resulting from RB loss may render cells sensitive to epigenetic perturbations of chromatin structure. RB loss sensitizes cells to high levels of both replication stress and DNA damage following PARP1/2 inhibition, and cells lacking RB are sensitive to the acquisition of replication-dependent DNA double strand breaks. The study also suggests that continued cell cycle progression cannot be maintained following both RB loss and PARP1/2 inhibition.

Point 1: RB-deficient cells are sensitive to PARP1 inhibition - strongly supportive

Point 2: Accumulation of DNA damage following RB loss and PARP inhibition is replication dependent - supportive

Point 3: Persistent replication in the presence of damage perpetuates genomic instability and compromises cell viability - very supportive

Expression of shRNA and RB in the systems used is induced by doxycycline. The effect of doxycycline appears to be uncontrolled for in these experiments. Similarly, the shRNA vector used for the doxycycline-inducible system is uncontrolled for using scramble shRNA or tet-Halo for the tet-RB-Halo.

To address these points, we have added additional negative controls to demonstrate that neither doxycycline, nor a GFP-Halo construct impacts the levels of DNA damage in these assays.

We now show that doxycycline-induced expression of an RB-targeting shRNA, but not doxycycline treatment alone, cooperates with PARP inhibition to promote replication stress and DNA damage in RPE cells (new panels Supplemental Figure S1C, D; S3A, B). Similarly, we have also added new data to show that doxycycline-induced expression of RB-Halo, but not GFP-Halo is sufficient to limit DNA damage following PARP inhibition (new panels Supplemental Figure S2D, E). Together with our original experiments showing that expression of an RB-targeting siRNA (but not a non-targeting, control siRNA) and CRISPR-induced deletion of the RB1 gene (but not the isogenic, RB1 proficient RPE cell line) cooperate with PARP trapping to promote

DNA damage, these data indicate that increased DNA damage and enhanced sensitivity of cells to PARP trapping is specific to RB loss and not an off-target effect of doxycycline or RNA interference.

"Together these data demonstrate that RB-deficient cells are generally sensitive to combined inhibition of PARP1 and PARP2" is technically inaccurate given the author's results showing that Veliparib failed to affect cell survival. What renders sensitivity to PARP "inhibitors" is the trapping of PARP rather than inhibition of PARP function. Please revise the statement.

Following with the previous comment, the discussion fails to address this observation. If PAR-dependent signaling is necessary to sustain fork stability, how do the authors explain the absence of effects upon Veliparib treatment?

We have revised the results and discussion sections to reflect that our data suggest PARP trapping, not merely inhibition of PARP activity, underlies the described sensitivity of RB deficient cells.

"Critically, re-introduction of RB... is sufficient to rescue DNA damage..." Figure 2F graph is missing the comparison between WT+Olaparib and tet-RB-Halo+Olaparib that is relevant for this statement.

We have revised the text to reflect that this is a partial rescue and also added statistical analysis in Figure 2F comparing the WT + Olaparib and the RB KO tet-RB-Halo + Olaparib conditions. The level of DNA damage in the RB KO cells that re-express RB-Halo is not statistically different following Olaparib treatment than that seen in the RB proficient wild-type cells.

The increased in EdU foci presented in Figure 5 could also be interpreted as RB-depleted cells having shorter G2 phase following PARP trapping, yet this was dismissed. Cell cycle analysis of these cells would strengthen the conclusion that the increased EdU foci are indicative of under-replicated DNA.

To address this concern, we have added two additional experimental approaches. First, we have repeated the analysis of persistent replication in G2 using a 30 min EdU pulse (New supplemental figure SF5A, B). Consistent with our initial experiment using a 2h EdU pulse (represented in main Figure 5A, B), this new experiment indicates that following Olaparib exposure, cells lacking RB continue to incorporate nucleotides just minutes before mitotic entry. The capacity of cells to form Edu+ foci in such a short time prior to mitotic entry argues against (but does not preclude) that our analysis is merely catching the very end of a normal S phase before cells progress through an abbreviated G2. Therefore, in a second new experiment we have used an RPE FUCCI system to perform live cell imaging and cell cycle analysis. We measured the duration of G2 in siScr and siRB treated cells with and without Olaparib treatment. In this new experiment we find that G2 is in fact extended (not shortened) in RB depleted, Olaparib treated cells when compared to G2 duration in either Olaparib treatment or

RB-depletion alone. This extended duration of G2 supports, together with continued nucleotide incorporation in the final 30 minutes prior to mitotic entry, support a model whereby RB-depleted, PARP-trapped cells enter G2/mitosis with under replicated DNA.

Minor:

Figure 1 legend: panel B indicates that control is in black and shRB in gray, but that is the color scheme for panel C. Panel B has different colors for each shRB repeat.

The figure legend has been updated

Several figure legends: "biological replicates" unless these are 5 distinct cell lines, this are experimental replicates.

We have replaced "biological replicates" to "experimental replicates" throughout the manuscript

Reviewer #2 (Comments to the Authors (Required)):

This research conducted by Zamalloa et al., sought to identify a mechanistic explanation to rationalize previous research that demonstrated loss of the tumor suppressor retinoblastoma (RB) compromises genomic stability, renders cells sensitive to inhibition of epigenetic modulators, and confers sensitivity to inhibition of poly ADP ribose polymerase (PARP) enzymes. Screens using inhibitors to epigenetic regulatory enzymes identified a group of inhibitors that selectively reduced viability of RB-depleted cells through targeting of PARP1/2. The authors were able to identify the specific mechanism of reduced viability through comparison of drug mechanisms related to PARP inhibition. Controlled knockdown and re-expression of RB in cells receiving Olaparib treatment identified PARP1/2 combined inhibition sensitizes RB-deficient cells to high levels of DNA damage, and re-expression of RB was sufficient to reduce DNA damage in response to PARP1/2 inhibition. Figure 3 experiments used quantification of Edu and PAR following pulse-labelling clearly identified RB loss promotes replication dependent PARylation. Combined with the findings of Figure 4, which identified DNA damage accumulation at sites of replicative stress following PARP1/2 inhibition and RB depletion, these two figures provide strong supporting evidence for their conclusion that accumulation of DNA damage following RB loss and PARP inhibition is replication dependent. The extended effects of persistent replication in the presence of this damage were also investigated and the researchers concluded an increase in under-replicated DNA, increased frequency of micronuclei, and continued cell cycle progression cannot be maintained following RB loss and PARP inhibition. Overall, this paper does well in connecting findings of previous research that identified RB and its role in genomic stability and how it can impact therapeutics through a specific and well-described mechanism. This paper provides clear mechanistic groundwork that would support more translational based investigations, for example, if these effects could be reproduced in various cancer cell lines and if there is synergistic or contraindicatory effects with other cancer therapeutics. However, upon review there are some potential revisions to be considered by these researchers.

1) The representative images of Figure 5C and their insets are difficult for the reader to identify the presence of micronuclei from the given images. The current magnification box partially obscures the identified cell in question. Recommend this be revised and that more levels of magnification, and some images without annotations to allow readers to see the micronuclei for themselves.

Figure 5C has been revised so that the magnification box no longer obscures the selected field of view and so that 'insets' now appear below the corresponding panel, allowing for larger, obstruction free representation of the micronuclei.

2) The second recommendation is the addition of a schematic summary in Figure 6 or as an additional figure. This is a largely mechanistic study, so providing a clear description of the postulated mechanism of how RB loss cooperates with PARP1/2 inhibition is key.

A schematic representing how the main conclusions of this study support our preferred model is now included in Figure 6.

3) The final recommendation would be review of minor editing in the figure legends and text. For example, the description of Figure 1B states this graph should be made of black and grey data points where it is made of four different colored data points.

We have reviewed and revised the text and figure legends to correct the descriptions and fix grammatical errors.

Referee Cross-Comments - The other reviewers have questions about the complementation experiment where RB is re-expressed in RB knock out cells. This is clearly a partial effect. However, compared to other attempts at this type of RB add back made over the years it is quite impressive. Likely because RB regulates so many different epigenetic mechanisms at different stages of the cell cycle, we've all discovered that you can't re-program cells back to normal with brief re-expression of RB. I see this experiment as a strength.

Reviewer #3 (Comments to the Authors (Required)):

The manuscript reports selectively increased sensitivity to PARP1,2 inhibitors, independently confirming reports from several preceding publications, but expanding these observations to an engineered retinal cell model. Interestingly, and notably, the authors show data that RB1 loss leads to increased PARylation in S-phase cells, raising the possibility that increased reliance on PARylation during S-phase explains the increased PARP inhibitor sensitivity of cells lacking RB.

Hence, the manuscript presents novel and attractive information. However, there are

shortcomings in this manuscript that the authors should seek to address. On many occasions, the authors seem to jump to conclusions and a balanced view of the data is not presented. Most significantly, selective sensitivity in their cell model is seen using Veliparib, albeit this PARP inhibitor does not show the described increase in pRPA and associated DNA damage phenotype, raising the question of whether these additional phenotypes, which are extensively studied in the work, are relevant for the increased sensitivity seen in their cell model.

The veliparib data call into question statements in the abstract and throughout the manuscript whereby "PARP inhibition" acts by causing replication-associated DNA damage. It is likely that many of the observations made are a consequence of the PARP complex trapping, and do not reflect PARP inhibition. How the observed responses relate to the selective sensitivity to these inhibitors is unclear. As a minimum these concerns need to be addressed by robustly rewriting the manuscript, detailing that the observed responses are not seen with Veliparib and ideally covering this discrepancy, and its implications, in the discussion. The abstract should be carefully examined. A number of statements presented are challenged by experimental data using Veliparib.

We have revised the abstract and the text to better reflect the presumed role of PARP trapping in our assays. We have expanded the discussion to suggest that robustness of the synergy we describe between RB loss and PARP trapping may require the physical impediment to replication that results from PARP trapping.

Other concerns that the authors should seek to address are:

There appear to be a number of unsupported statements throughout the manuscript.

For example:

The introduction states: "we find that inhibition of PARP activity permits acceleration of replication fork progression" albeit no experiments are shown that measure fork progression.

We have removed this statement and revised the introduction to more accurately depict the data presented in this manuscript.

The result section states: "RB1 loss sensitises cells to high level of both replication stress" ... while in fact, the data show no difference in pRPA (a common readout for replication stress) between RB-containing and RB-depleted cells (figure 4B, supplemental figure 3B supplemental figure 4D). Hence at least using this canonical assay for replication stress there is no evidence supporting the statement made.

The intensity of pRPA staining in RB-depleted cells treated with PARP trappers is sufficiently high (and the corresponding exposure time of images captured sufficiently short) that it is not possible to discern differences in pRPA levels between control and RB-depleted cells in this experiment. Therefore, to better assess levels of replication stress following RB depletion, we pulse labelled control and shRB cells with EdU and stained for pRPA. Nuclei were identified using DAPI staining and a threshold set for EdU positivity so that cells could be assigned as either EdU negative or EdU positive. The average nuclear pRPA intensity was determined for the

EdU negative control cells and the frequency of cells in the EdU+ and EdU- populations with 2-fold or greater nuclear pRPA intensity was calculated for each condition. Consistent with previous reports demonstrating that RB depletion promotes replication stress, these analyses reveal the frequency of pRPA-positive S phase cells doubles when RB is depleted. These new data are represented in Supplemental Figure S3A.

The discussion states that: "We show .. that when PARP1/2 is inhibited sites of replication stress accumulate DNA damage during S phase...". This statement is contradicted by data using Veliparib, which inhibits PARP 1/2 but does not cause either of these phenotypes that:

We have revised this statement (and similar statements throughout the text) to reflect that the effect is seen with PARP trapping, not PARP inhibition *per se*.

further down is a statement: "PARP is activated by single strand DNA, parylating various substrates to stabilize the replication fork (Yang et al 2004)", yet the reference cited does not report any such data.

statistics use:

The method section states that unpaired t-tests were used to assess the significance of variance. However, this is not meaningful and appropriate for some data. Specifically, a 2-way ANOVA scoring for "variables interaction" should be used to assess the ability of RB re-expression to rescue the DNA damage following Olaparib. The same applies to data in Figure 6B assessing if Olaparib "interacts" with RB1 loss to decrease EDU-positive cells.

We have corrected our statistical analysis and now use a 2-way ANOVA to score for variable interaction for panels in figures 2F and 6B. This new analysis does not change the interpretation of the data and the new statistical approach is reflected in the revised figures, legends, and methods.

Minor :

The concentration of all drugs used should be detailed in the respective figure legends.

We have updated the figure legends to include details of the drug concentrations used in the respective experiments.

Data volume statements, please can it be clearly spelled out for each dataset if biological repeats for experiments were run in parallel, or were independent.

All replicates were performed independently. We have updated the methods to clarify this.

August 29, 2023

RE: Life Science Alliance Manuscript #LSA-2023-02067-TR

Dr. Amity Manning
Worcester Polytechnic Institute
Biology and Biotechnology
60 Prescott St
Worcester, MA 01605

Dear Dr. Manning,

Thank you for submitting your revised manuscript entitled "RB loss sensitizes cells to replication-associated DNA damage following PARP inhibition by trapping". We would be happy to publish your paper in Life Science Alliance pending final revisions necessary to meet our formatting guidelines.

- please address Reviewer 1's remaining comments
- please add your main and supplementary figure legends to the main manuscript text after the references section
- please make sure the author order in your manuscript and our system match; the full name (middle names as initials) of each author should be given on the title page
- we encourage you to revise the figure legend for Figure 6 such that the figure panels are introduced in an alphabetical order
- please indicate sizes next to each blot

A. FINAL FILES:

B. MANUSCRIPT ORGANIZATION AND FORMATTING:

**Submission of a paper that does not conform to Life Science Alliance guidelines will delay the acceptance of your

manuscript.**

The license to publish form must be signed before your manuscript can be sent to production. A link to the electronic license to publish form will be sent to the corresponding author only. Please take a moment to check your funder requirements.

Sincerely,

Reviewer #1 (Comments to the Authors (Required)):

The authors have appropriately addressed the questions raised by the initial review. The manuscript should be suitable for publication with a few last edits.

Minor revision to make:

1. Running title should change PARP "inhibition" to "trapping"
2. Figure Legend 6 is missing the description for panels H and I.

Reviewer #2 (Comments to the Authors (Required)):

I'm satisfied that all concerns have been addressed.

Reviewer #3 (Comments to the Authors (Required)):

The manuscript examines the molecular response to PARP inhibitors in cells with loss of Retinoblastoma suppressor protein. The authors in their revised version responded fully to all concerns raised in my review of their initial submission.

This is carefully conducted work providing novel insight into the cellular consequences of retinoblastoma protein loss in cancer with added implications for the therapy in this cancer group.

The reviewer thanks the authors for their response which fully satisfies and constructively addresses any concerns raised.

I fully support the acceptance of this manuscript for publication in its resubmitted form.

September 6, 2023

RE: Life Science Alliance Manuscript #LSA-2023-02067-TRR

Dr. Amity Manning
Worcester Polytechnic Institute
Biology and Biotechnology
60 Prescott St
Worcester, MA 01605

Dear Dr. Manning,

Thank you for submitting your Research Article entitled "RB loss sensitizes cells to replication-associated DNA damage following PARP inhibition by trapping". It is a pleasure to let you know that your manuscript is now accepted for publication in Life Science Alliance. Congratulations on this interesting work.

DISTRIBUTION OF MATERIALS:

Again, congratulations on a very nice paper. I hope you found the review process to be constructive and are pleased with how the manuscript was handled editorially. We look forward to future exciting submissions from your lab.

Sincerely,
